# Mathematical modeling of the West Africa Ebola epidemic

**Jean-Paul Chretien[1]\*, Steven Riley[2], Dylan B George[3]**

[1]Department of Defense, Division of Integrated Biosurveillance, Armed Forces Health Surveillance Center, Silver Spring, United States; [2]MRC Centre for Outbreak Analysis and Modelling, Department of Infectious Disease Epidemiology, School of Public Health, Imperial College London, London, United Kingdom; [3]Department of Health and Human Services, Biomedical Advanced Research and Development Authority, Washington, United States

**Abstract** As of November 2015, the Ebola virus disease (EVD) epidemic that began in West Africa in late 2013 is waning. The human toll includes more than 28,000 EVD cases and 11,000 deaths in Guinea, Liberia, and Sierra Leone, the most heavily-affected countries. We reviewed 66 mathematical modeling studies of the EVD epidemic published in the peer-reviewed literature to assess the key uncertainties models addressed, data used for modeling, public sharing of data and results, and model performance. Based on the review, we suggest steps to improve the use of modeling in future public health emergencies.

**\*For correspondence:**
JPChretien@gmail.com

**Competing interests:** The authors declare that no competing interests exist.

## Introduction

On March 23, 2014, the Ministry of Health Guinea notified the World Health Organization (WHO) of a rapidly evolving outbreak of Ebola virus disease (EVD), now believed to have begun in December 2013. The epidemic spread through West Africa and reached Europe and the United States. As of November 4, 2015, WHO reported more than 28,000 cumulative cases and 11,000 deaths in Guinea, Liberia, and Sierra Leone, where transmission had been most intense (*World Health Organization, 2016*).

As the emergency progressed, researchers developed mathematical models of the epidemiological dynamics. Modelers have assessed ongoing epidemics previously, but the prominence of recent EVD work, enabled by existing research programs for infectious disease modeling (*National Institutes of Health, 2016a*; *National Institutes of Health, 2016b*) and online availability of EVD data via WHO (*World Health Organization, 2016*), Ministries of Health of affected countries, or modelers who transcribed and organized public WHO or Ministry of Health data (*Rivers C*) may be unprecedented. The efforts for this outbreak have been numerous and diverse, with major media incorporating modeling results in many pieces throughout the outbreak. U.S. Government decision making has benefited from modeling results at key moments during the response (*Robinson R*).

We draw on this vigorous response of the epidemiological modeling community to the EVD epidemic to review (*Moher et al., 2009*) the application of modeling to public health emergencies, and identify lessons to guide the modeling response to future emergencies.

## Results

### Overview of modeling applications
We identified 66 publications meeting inclusion criteria (*Figure 1*).

**eLife digest** The outbreak of Ebola that started in West Africa in late 2013 has caused at least 28,000 illnesses and 11,000 deaths. As the outbreak progressed, global and local public health authorities scrambled to contain the spread of the disease by isolating those who were ill, putting in place infection control processes in health care settings, and encouraging the public to take steps to prevent the spread of the illness in the community. It took a massive investment of resources and personnel from many countries to eventually bring the outbreak under control.

To determine where to allocate people and resources during the outbreak, public health authorities often turned to mathematical models created by scientists to predict the course of the outbreak and identify interventions that could be effective. Many groups of scientists created models of the epidemic using publically available data or data they obtained from government officials or field studies. In some instances, the models yielded valuable insights. But with various groups using different methods and data, the models didn't always agree on what would happen next or how best to contain the epidemic.

Now, Chretien et al. provide an overview of Ebola mathematical modeling during the epidemic and suggest how future efforts may be improved. The overview included 66 published studies about Ebola outbreak models. Although most forecasts predicted many more cases than actually occurred, some modeling approaches produced more accurate predictions, and several models yielded valuable insights. For example, one study found that focusing efforts on isolating patients with the most severe cases of Ebola would help end the epidemic by substantially reducing the number of new infections. Another study used real-time airline data to predict which traveler screening strategies would be most efficient at preventing international spread of Ebola. Furthermore, studies that obtained genomic data showed how specific virus strains were transmitted across geographic areas.

Chretien et al. argue that mathematical modeling efforts could be more useful in future pubic health emergencies if modelers cooperated more, and suggest the collaborative approach of weather forecasters as a good example to follow. Greater data sharing and the creation of standards for epidemic modeling would aid better collaboration.

Models addressed 6 key uncertainties about the EVD epidemic: transmissibility, typically represented by the reproduction number ($R$, the average number of people each infected person infects; assessed in 41 publications); effectiveness of various interventions that had been or might be implemented (in 29 publications); epidemic forecast (in 29 publications); regional or international spreading patterns or risk (in 15 publications); phylogenetics of EVD viruses (in 9 publications); and feasibility of conducting vaccine trials in West Africa (in 2 publications) (*Table 1*, *Supplementary file 1*).

The number of publications with models to estimate $R$ increased rapidly early in the epidemic, along with those including intervention, forecasting, and regional and international spread models; the growth rate of publications with phylogenetic modeling applications and clinical trial models increased later in the epidemic (*Figure 2*).

Of the 125 models reported across the studies, 74% included mechanistic assumptions about disease transmission (e.g., compartmental, agent-based, or phylogenetic models), while 26% were purely phenomenological (*Supplementary file 2*).

## Data sources

For 54 (82%) of the 66 publications, the only EVD data used was pre-existing and publicly-available (*Table 1*). Typically, these were aggregate case data posted online by the WHO or affected countries, or Ebola virus genetic data released previously during the epidemic. Twelve studies used original EVD epidemiological data (*Baize et al., 2014*; *WHO Ebola Response Team, 2014*; *2015*; *Faye et al., 2015*; *Yamin et al., 2014*) or genomic data (*Baize et al., 2014*; *Gire et al., 2014*; *Simon-Loriere et al., 2015*; *Tong et al., 2015*; *Hoenen et al., 2015*; *Park et al., 2015*; *Carroll et al., 2015*; *Kugelman et al., 2015*).

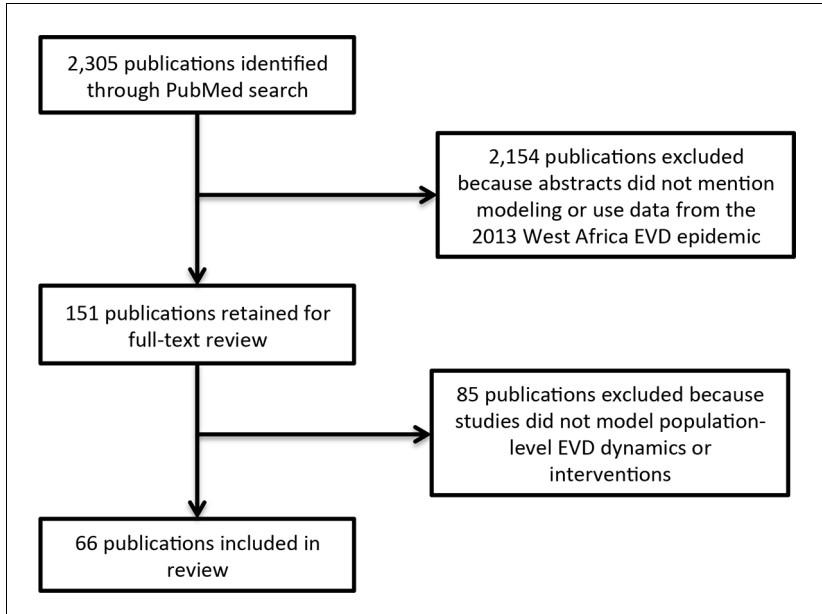

**Figure 1.** Literature search flow.

Examples of additional data used for some modeling applications include official reports of social mobilization efforts (*Fast et al., 2015*), media reports of case clusters (*Cleaton et al., 2015*), media reports of events that may curtail or aggravate transmission (*Majumder et al., 2014*), and international air travel data (*Gomes et al., 2014*; *Poletto et al., 2014*; *Read et al., 2015*; *Bogoch et al., 2015*; *Rainisch et al., 2015*; *Cope et al., 2014*). Several studies incorporated spatial data on EVD cases into models of regional EVD spread (*Gire et al., 2014*; *Merler et al., 2015*; *Rainisch et al., 2015*; *Tong et al., 2015*; *Carroll et al., 2015*; *Zinszer et al., 2015*).

## Data and results sharing

Of the 12 studies that collected original EVD data, 9 released those data before or at the time of publication (8 with Ebola virus genetic data deposited in GenBank (*Baize et al., 2014*; *Gire et al., 2014*; *Simon-Loriere et al., 2015*; *Tong et al., 2015*; *Hoenen et al., 2015*; *Park et al., 2015*; *Carroll et al., 2015*; *Kugelman et al., 2015*) and 1 with detailed epidemiological data in the online publication (*Yamin et al., 2014*). Many publications used results from the WHO Ebola Response Team investigations (*WHO Ebola Response Team, 2014*; *2015*) (for example, estimates of the generation time, case fatality rate, or other epidemiological parameters as model inputs), but the detailed epidemiological data from these studies, to date, are not publicly available.

Accumulation of shared EVD data over successive studies was evident especially in the phylogenetic analyses. For example, all phylogenetic studies published after release of the initial Ebola virus sequences by (*Baize et al., 2014*) (Guinea) and (*Gire et al., 2014*) (Sierra Leone) incorporated those sequence data.

Across all studies, the publication lag (defined as date of most recent EVD data used to date of online publication) was almost 3 months (median [interquartile range] = 85 [30–157] days). The lag varied across modeling applications, and was considerably shorter in studies that included models to estimate $R$ (median = 58 days for publications with $R$ estimation versus 118 days for others) or to forecast (median = 50 versus 125 days) (*Figure 3*).

Lags were longest for studies with phylogenetic and clinical trials applications (median = 125 and 108 days, respectively), although there were fewer publications with these models.

**Table 1.** Overview of modeling publications on the 2013-present EVD epidemic.

| Ref. | Date of latest EVD data | Date published | EVD data was pre-existing and public | Uncertainties addressed | | | | | |
|------|-------------------------|----------------|--------------------------------------|---|---------------|----------|--------|---------------|-------------------|
|      |                         |                |                                      | R | Interventions | Forecast | Spread | Phylogenetics | Clinical trials |
| Baize et al., 2014 | 3/20/14 | 4/16/14 | No | | | | | * | |
| Dudas and Rambaut, 2014 | 3/20/14 | 5/2/14 | Yes | | | | | * | |
| Alizon et al., 2014 | 6/18/14 | 12/13/14 | Yes | * | | * | | | |
| Gire et al., 2014 | 6/18/14 | 8/28/14 | No | | | | * | * | |
| Stadler et al., 2014 | 6/18/14 | 10/6/14 | Yes | * | | | | | |
| Volz and Pond, 2014 | 6/18/14 | 10/24/14 | Yes | * | | | | | |
| Pandey et al., 2014 | 8/7/14 | 10/30/14 | Yes | * | * | * | | | |
| Gomes et al., 2014 | 8/9/14 | 9/2/14 | Yes | * | | * | * | | |
| Valdez et al., 2015 | 8/15/14 | 7/20/15 | Yes | * | * | * | * | | |
| Merler et al., 2015 | 8/16/14 | 1/7/15 | Yes | * | * | * | * | | |
| Rainisch et al., 2015 | 8/16/14 | 2/18/15 | Yes | | | | * | | |
| Althaus, 2014 | 8/20/14 | 9/2/14 | Yes | * | | | | | |
| Fisman et al., 2014 | 8/22/14 | 9/8/14 | Yes | * | | * | | | |
| Nishiura and Chowell, 2014 | 8/26/14 | 9/11/14 | Yes | * | | * | | | |
| Poletto et al., 2014 | 8/27/14 | 10/23/14 | Yes | | * | | * | | |
| Meltzer et al., 2014 | 8/28/14 | 9/26/14 | Yes | * | * | * | | | |
| Agusto et al., 2015 | 8/29/14 | 4/23/15 | Yes | * | * | | | | |
| Althaus, 2015 | 8/31/14 | 4/19/15 | Yes | * | | | | | |
| Scarpino et al., 2014 | 8/31/14 | 12/15/14 | Yes | * | | | | | |
| Weitz and Dushoff, 2015 | 8/31/14 | 3/4/15 | Yes | * | * | | | | |
| Drake et al., 2015 | 9/2/14 | 10/30/14 | Yes | * | * | * | | | |
| Towers et al., 2014 | 9/8/14 | 9/18/14 | Yes | * | | * | | | |
| Bellan et al., 2014 | 9/14/14 | 10/14/14 | Yes | | | * | | | |
| Chowell et al., 2015 | 9/14/14 | 1/19/15 | Yes | | * | | | | |
| Cooper et al., 2015 | 9/14/14 | 4/14/15 | Yes | | | | | | * |
| Read et al., 2015 | 9/14/14 | 11/12/14 | Yes | | * | | * | | |
| WHO Ebola Response Team, 2014 | 9/14/14 | 9/23/14 | No | * | | * | * | | |
| Faye et al., 2015 | 9/16/14 | 1/23/15 | No | * | * | | | | |
| Bogoch et al., 2015 | 9/21/14 | 10/21/14 | Yes | | * | | * | | |
| Yamin et al., 2014 | 9/22/14 | 10/28/14 | No | * | * | | | | |
| Lewnard et al., 2014 | 9/23/14 | 10/24/14 | Yes | * | * | * | | | |
| Webb et al., 2015 | 9/23/14 | 1/30/15 | Yes | * | * | * | | | |
| Shaman et al., 2014 | 9/28/14 | 10/27/14 | Yes | * | | * | | | |
| Chowell et al., 2014 | 10/1/14 | 11/20/14 | Yes | * | | * | | | |
| Fasina et al., 2014 | 10/1/14 | 10/9/14 | Yes | | * | | | | |
| Khan et al., 2015 | 10/1/14 | 2/24/15 | Yes | * | | | | | |
| Rivers et al., 2014 | 10/5/14 | 10/16/14 | Yes | * | * | * | | | |
| Xia et al., 2015 | 10/7/14 | 9/8/15 | Yes | * | * | | | | |
| Majumder et al., 2014 | 10/11/14 | 4/28/15 | Yes | * | * | | | | |

*Table 1 continued on next page*

Chretien *et al*. eLife 2015;4:e09186. DOI: 10.7554/eLife.09186

*Table 1 continued*

| Ref. | Date of latest EVD data | Date published | EVD data was pre-existing and public | Uncertainties addressed | | | | | |
| --- | --- | --- | --- | --- | --- | --- | --- | --- | --- |
| | | | | R | Interventions | Forecast | Spread | Phylogenetics | Clinical trials |
| Kiskowski, 2014 | 10/15/14 | 11/13/14 | Yes | * | | * | | | |
| Fisman and Tuite, 2014 | 10/18/14 | 11/21/14 | Yes | * | * | * | | | |
| Althaus et al., 2015 | 10/20/14 | 1/15/15 | Yes | * | * | * | | | |
| Simon-Loriere et al., 2015 | 10/25/14 | 6/24/15 | No | | | | | * | |
| Rainisch et al., 2015 | 10/31/14 | 6/16/15 | Yes | | | | * | | |
| Fast et al., 2015 | 11/1/14 | 5/15/15 | Yes | | * | | | | |
| Kucharski et al., 2015 | 11/1/14 | 2/18/15 | Yes | * | * | * | | | |
| Tong et al., 2015 | 11/11/14 | 5/13/15 | No | | | | * | * | |
| Hoenen et al., 2015 | 11/21/14 | 3/26/15 | No | | | | | * | |
| Cope et al., 2014 | 12/3/14 | 12/10/14 | Yes | | * | | * | | |
| White et al., 2014 | 12/3/14 | 1/30/15 | Yes | * | * | * | | | |
| WHO Ebola Response Team, 2015 | 12/14/14 | 12/24/14 | No | * | | * | | | |
| Chowell et al., 2014 | 12/17/14 | 1/21/15 | Yes | * | | | | | |
| Siettos et al., 2014 | 12/21/14 | 3/9/15 | Yes | * | | * | | | |
| Park et al., 2015 | 12/26/14 | 6/18/15 | No | | | | | * | |
| Nadhem and Nejib, 2015 | 12/30/14 | 6/14/15 | Yes | | | * | | | |
| Camacho et al., 2015 | 1/18/15 | 2/10/15 | Yes | * | | * | | | |
| Carroll et al., 2015 | 1/31/15 | 6/17/15 | No | | | | * | * | |
| Bellan et al., 2015 | 2/9/15 | 4/15/15 | Yes | | | * | | | * |
| Barbarossa et al., 2015 | 2/13/15 | 7/21/15 | Yes | * | * | * | | | |
| Kugelman et al., 2015 | 2/14/15 | 6/12/15 | No | | | | | * | |
| Cleaton et al., 2015 | 2/28/15 | 9/3/15 | Yes | * | | | | | |
| Wang and Zhong, 2015 | 3/18/15 | 3/24/15 | Yes | * | | | | | |
| Toth et al., 2015 | 3/31/15 | 7/14/15 | Yes | | * | | * | | |
| Dong et al., 2015 | 4/3/15 | 9/5/15 | Yes | | * | * | | | |
| Browne et al., 2015 | 4/12/15 | 5/14/15 | Yes | * | * | | | | |
| Zinszer et al., 2015 | 5/13/15 | 9/1/15 | Yes | | | | * | | |

## Modeling results: *R* and forecasts

Forty-one publications characterized epidemic dynamics using epidemiological (N=36), genomic (N=4), or news report data (N=1). Twenty-four of these provided estimates of the basic reproduction number ($R_0$) for Guinea, Liberia, Sierra Leone, or West Africa, using epidemiological or genomic data (*Figure 4*, *Supplementary file 3*).

There were 16 country-specific estimates of $R_0$ for Guinea, Liberia, or Sierra Leone that used EVD epidemiological data (aggregate or line-level) and provided 95% confidence or credible intervals

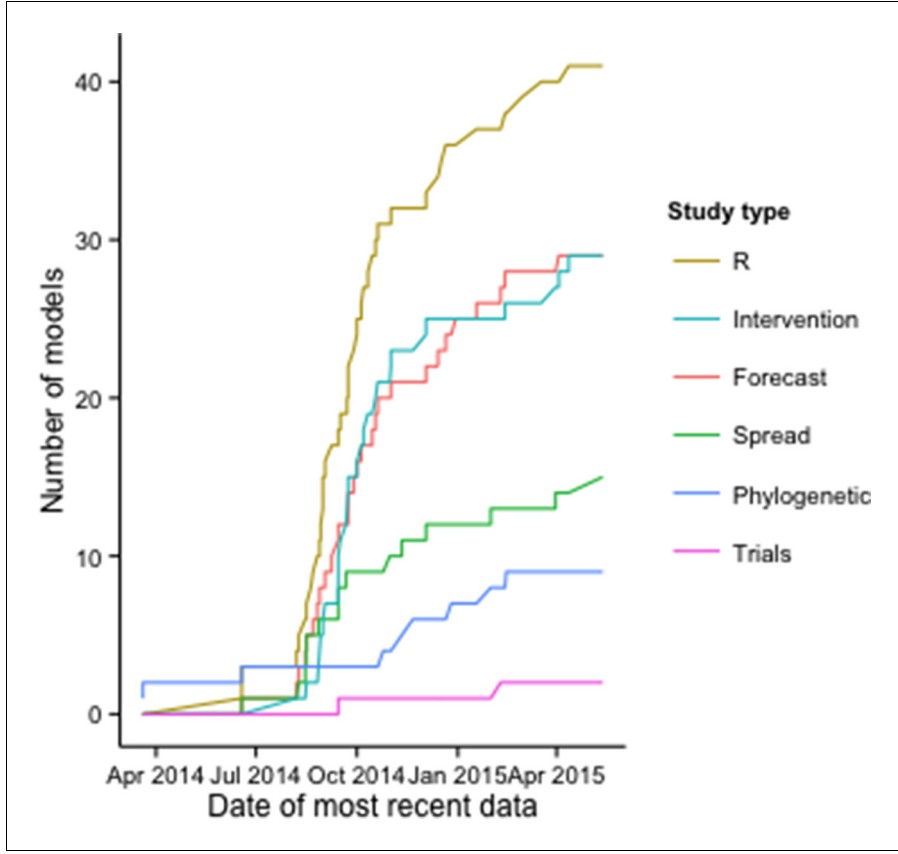

**Figure 2.** Cumulative number of modeling applications by date of most recent EVD data used. The figure includes 125 modeling applications across the 66 publications.

(CIs). Median CI width was about 85% smaller for models that used cumulative EVD counts (N=11 models in 5 publications) than for models that used disaggregated EVD case data, such as weekly counts (N=5 models in 3 publications) (*Figure 5*).

Although CIs were also narrower for models when deterministic rather than stochastic methods were used to estimate parameter uncertainty, all of the deterministic results came from a single study (*Figure 6*).

Fifteen publications provided numerical forecasts of cumulative EVD incidence for West African countries. Of 22 models that assumed no additional response measures beyond those implemented at the time (i.e., 'status quo' assumptions), 18 overestimated the future number of cases (*Figure 7*, *Supplementary file 4*).

In multivariate analysis, forecast error was lower for forecasts made later in the outbreak (14% reduction in mean absolute percentage error [MAPE] per week, P<0.001), higher for forecasts with longer time horizons (29% increase in MAPE per week, P<0.01), and lower for forecasts that used decay terms, spatially heterogeneous contact patterns, or other methods that served to constrain projected incidence growth (90% reduction in MAPE, P<0.01). Country and number of parameters in the model were not statistically significant predictors of forecast accuracy.

## Discussion

We identified 66 modeling publications during approximately 18 months of the EVD response that assessed trends in the intensity of transmission, effectiveness of control measures, future case counts, regional and international spreading risk, Ebola virus phylogenetic relationships and recent evolutionary dynamics, and feasibility of clinical trials in West Africa. We found a heavy dependence

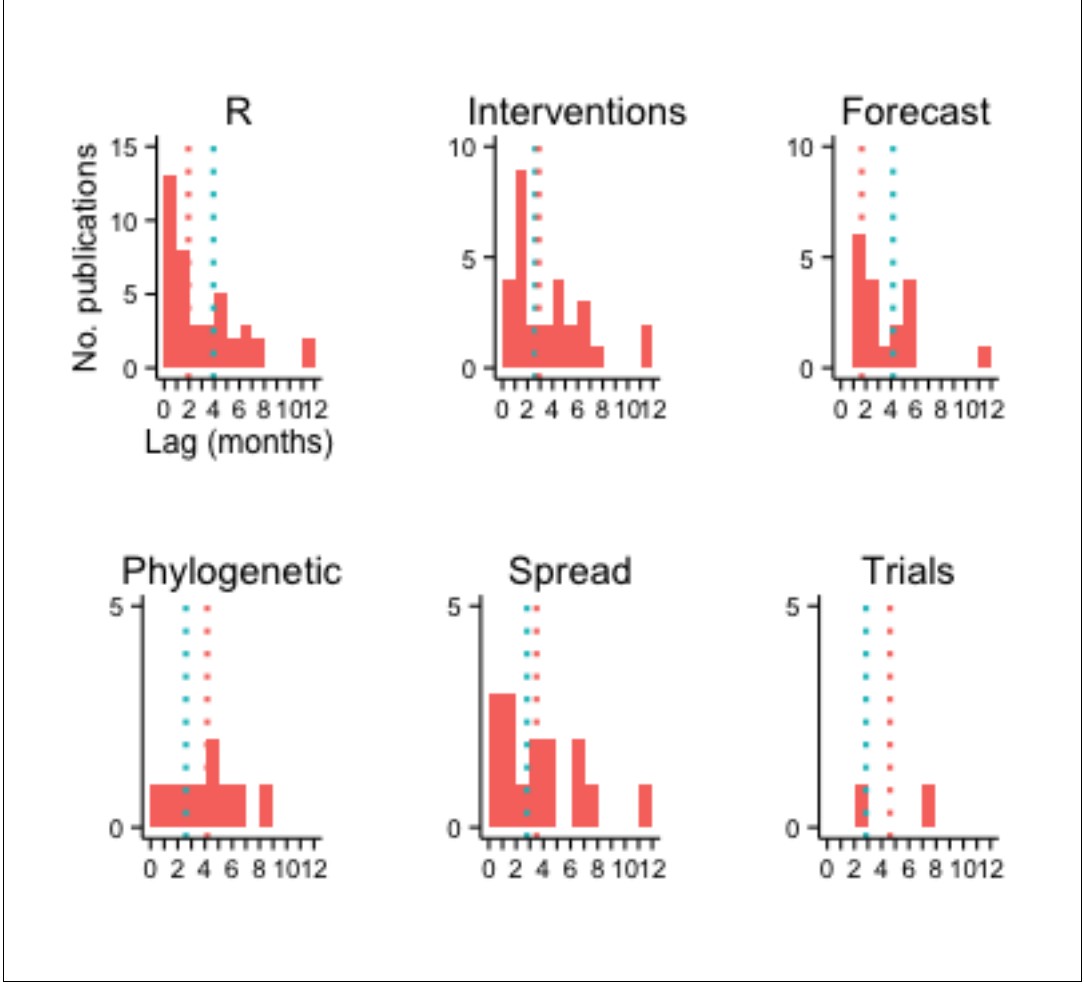

**Figure 3.** Publication lag by type of modeling application. The vertical red and turquoise lines indicate the median lag for publications including and not including, respectively, the type of modeling application.

on public data for EVD modeling, and identified factors that might have influenced model performance. To our knowledge, this review is one of the most comprehensive assessments of mathematical modeling applied to a single real-world public health emergency.

An important caveat of our review is that it only captures published results. We are aware of additional EVD epidemiological investigations and modeling not yet published. Some modelers providing direct support to operational response efforts have not published results, possibly because of operational demands.

Also, we could not account comprehensively for the sources of variation across studies. For example, studies that estimated $R_0$ using the same data sources at about the same time reported varied results. Such variation may, in part, reflect the problem of identifiability, with different $R_0$ estimates possible for models that perform equally well depending on other parameter values (*Weitz and Dushoff, 2015*). Ideally, an investigation into this heterogeneity would include implementation of models in a common testing environment.

Our review suggests several possible steps for improving the application of epidemiological modeling during public health emergencies. First, agreement on community best practices could improve the quality of modeling support to decision-makers. For example, our analysis is consistent with simulation studies showing underestimation of uncertainty in estimating $R_0$ with cumulative (as opposed to disaggregated) incidence data, and supports the recommendation to use disaggregated

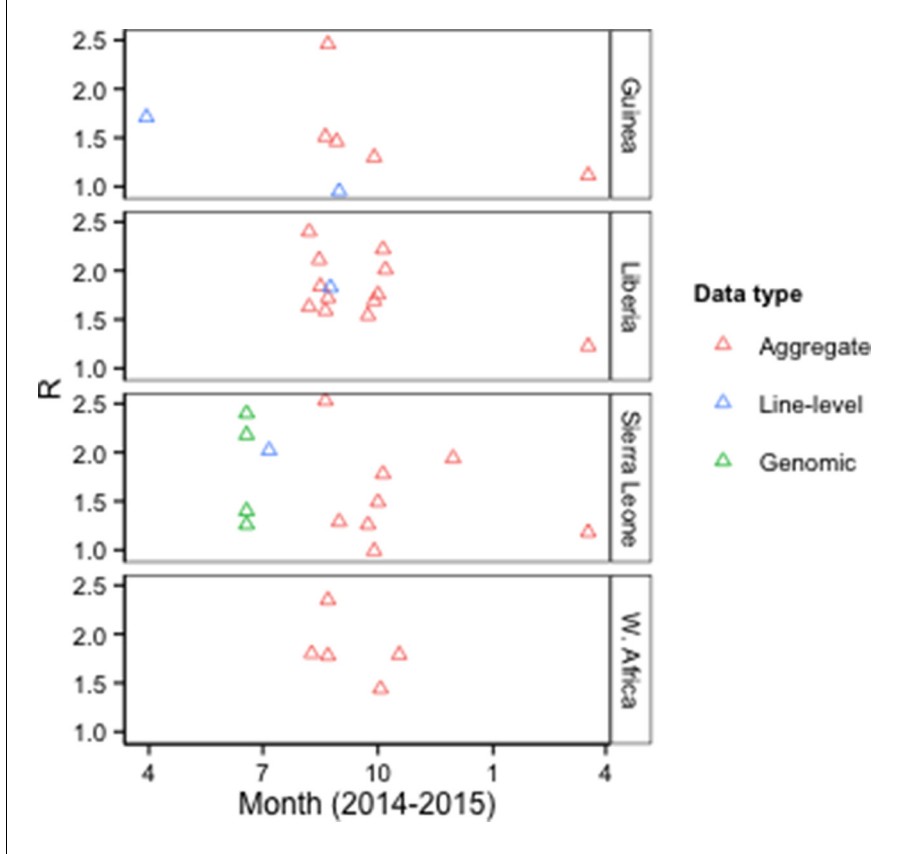

**Figure 4.** $R_0$ estimates by type of model input data. Aggregate, case counts released by the WHO or Ministries of Health; Line-level, individual-level data from epidemiological investigations; Genomic, Ebola virus sequence data. The Figure excludes an outlier estimate of 8.33 for Sierra Leone (*Fisman et al., 2014*).

data and stochastic models (*King et al., 2015*). Additionally, incidence forecasts provided reasonable prospective estimates several weeks forward in time during the initial phase; however, given available data and methodologies these forecasts became progressively more inaccurate as they projected dynamics beyond several weeks. Validation of incidence forecasts against other relevant data, such as hospital admissions and contacts identified, also could provide evidence that the assumptions are sound.

The 2014 onwards ebola outbreak in West Africa clearly highlights the need for a better understanding of how increasing awareness of severe infections within a community decreases their transmissibility even in the absence of specific interventions. Advancing methodological approaches to capture this effect, such as dampening approaches, might help account for behavioral changes, interventions, contact heterogeneity, or other factors that can be expected in a public health emergency which likely will improve forecasting accuracy. Establishing best practices within the community will allow decision-makers the ability to more quickly accept methodologies and results that have been generated via these best practices. Hence, decisions based on these results can happen more quickly.

Second, modeling coordination could facilitate direct comparison of modeling results, identifying issues on which diverse approaches agree and areas of greater uncertainty. Epidemiological modelers might learn from comparison initiatives in modeling of influenza (*Centers for Disease Control and Prevention, 2013*), dengue (*US Department of Commerce*), and HIV (*HIV modeling consortium*); and in other fields such as climate forecasting *Intergovernmental Panel on Climate Change, 2010*). For epidemiological application, an ensemble approach should preserve methodological diversity to exploit the full range of state-

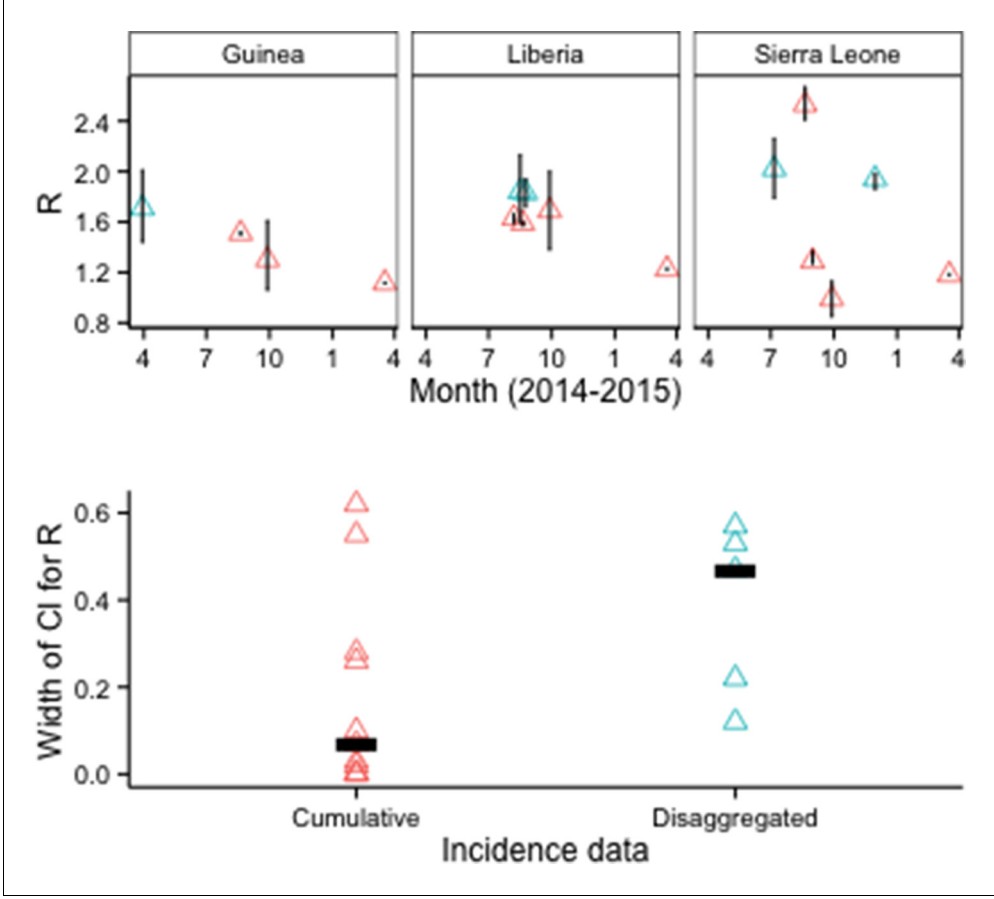

**Figure 5.** $R_0$ estimates and CIs by type of epidemiological input data. Disaggregated data typically were weekly counts. Top row: Vertical lines indicate 95% CIs. Bottom row: Horizontal bars indicate median CI width.

of-the-art modeling methods, but include enough standardization to enable cross-model comparison. Establishing an initial architecture for a coordinated, ensemble effort now could assist the response to EVD, and future public health emergencies.

Perhaps most importantly, outbreak modeling efforts would be much more fruitful if data and analytical results could be made available more quickly to all interested parties (*Yozwiak et al., 2015*). The publication timelines for academic journals typically will not be consistent with decision-making needs during public health emergencies like the EVD epidemic, where the epidemiological situation was highly dynamic and the usefulness of data and forecasts time-constrained. Establishing mechanisms for modelers without special access to the official epidemiological teams to share interim results would expand the evidence base for response decision-making. Ideally, data should be made available online in machine-readable form to facilitate use in analyses. Modelers and other analysts expended enormous effort during the EVD epidemic transcribing data posted online in pdf documents.

New norms for data-sharing during public health emergencies (*World Health Organization, 2015*) would remove the most obvious hurdle for model comparison. The current situation where groups either negotiate bilaterally with individual countries or work exclusively with global health and development agencies is understandable, but highly ineffective. The EVD outbreak highlights again – after the 2003 Severe Acute Respiratory Syndrome epidemic and 2009 influenza A (H1N1) pandemic – that an independent, well-resourced global data observatory could greatly facilitate the public health response in many ways, not least of which would be the enablement of rapid, high quality, and easily comparable disease-dynamic studies.

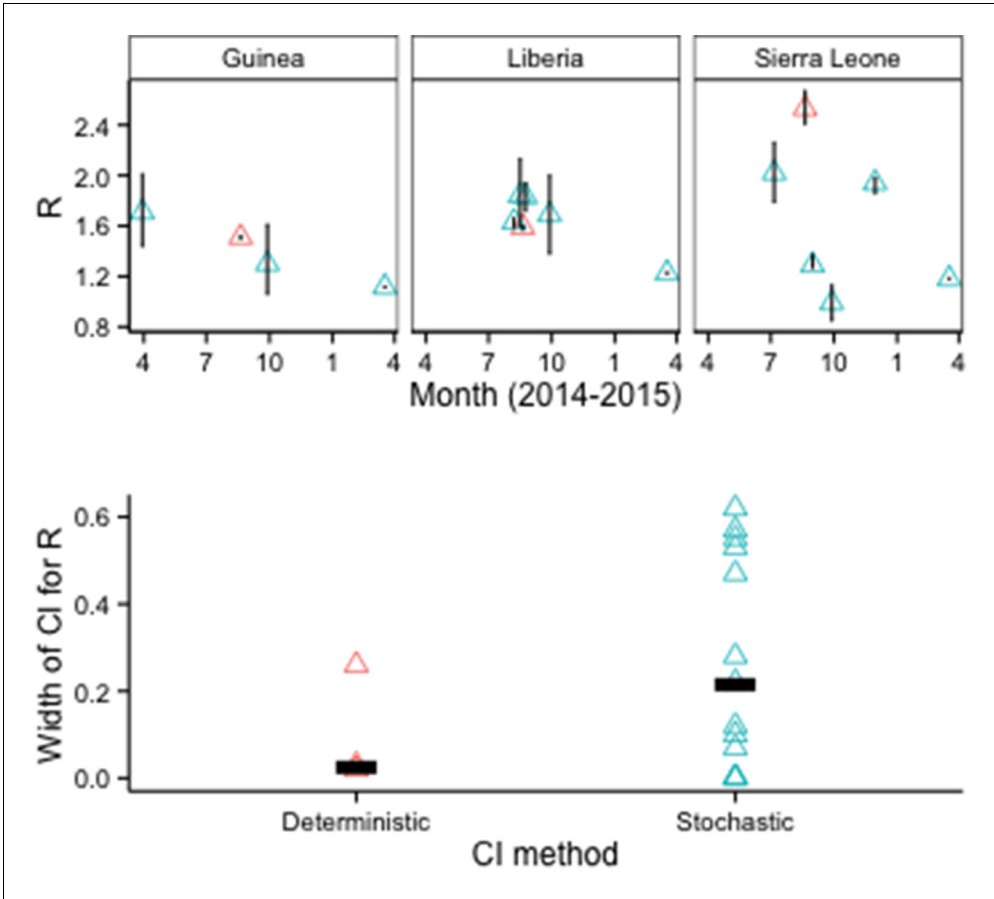

**Figure 6.** $R_0$ estimates and CIs by model fitting method. Top row: Vertical lines indicate 95% CIs. Bottom row: Horizontal bars indicate median CI width.

## Materials and methods

For this review, we adapted the PRISMA methodology (*Moher et al., 2009*) to identify quantitative modeling studies of the 2013-present West Africa EVD epidemic. We searched PubMed on September 24, 2015, for publications in English since December 1, 2013, using the term 'Ebola' in any field. We reviewed all returned abstracts and selected ones for confirmatory, full-text review that mentioned use of quantitative models to characterize or predict epidemic dynamics or evaluate interventions. We included studies that met this criterion in full-text review.

We excluded studies of clinical prediction models, viral or physiological function models, ecological niche models, animal reservoir models, and publications that did not use data from the 2013-present West Africa EVD epidemic.

For included publications, we recorded the geographic settings, date of most recent EVD data used and date of publication, type of EVD data used, questions the models addressed, modeling approaches, and key results, including estimates of the basic reproduction number ($R_0$) and forecasts of future EVD incidence provided in the main text of the publications. To assess forecast accuracy, we compared predictions of models made under 'status quo' assumptions (i.e., without explicit inclusion of additional interventions or behavioral changes) to EVD incidence data subsequently released by the WHO (*World Health Organization, 2016*), using the WHO figures dated soonest after the forecast target date.

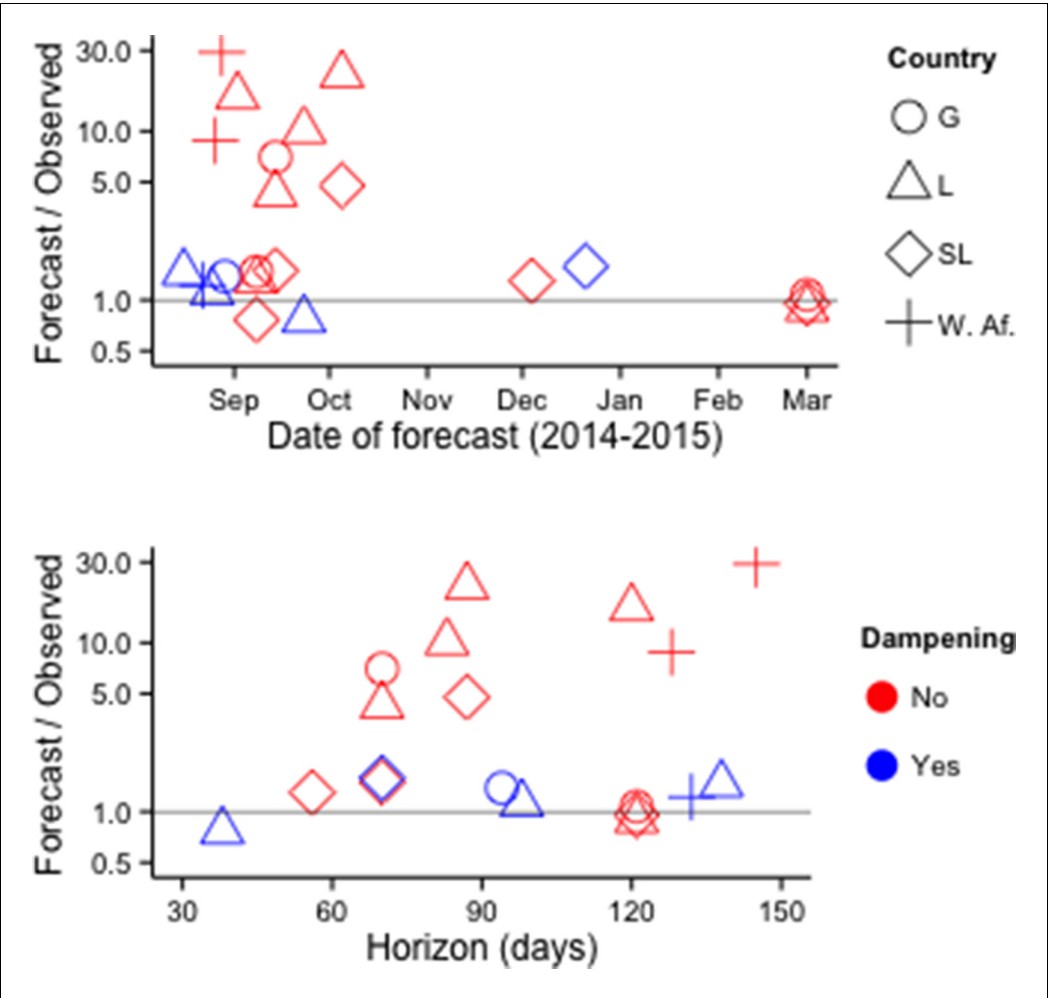

**Figure 7.** Accuracy of cumulative incidence forecasts. Accuracy is shown as the ratio of predicted incidence to incidence subsequently reported by the WHO. 'Dampening' refers to various approaches to restrict the growth of forecasted incidence over time. Top row: Accuracy by date of forecast. Bottom row: Accuracy by forecast lead time ('Horizon'). The Figure excludes one forecast with horizon > 1 year (*Fisman and Tuite, 2014*).

## Acknowledgements

We thank the reviewers for excellent comments that improved the manuscript. The views expressed are those of the authors and do not necessarily reflect the views of any part of the US Government.

## Additional information

### Funding
No external funding was received for this work.

### Author contributions
J-PC, SR, DBG, Conception and design, Acquisition of data, Analysis and interpretation of data, Drafting or revising the article

## Additional files

**Supplementary files**

• Supplementary file 1. Overview of publications.

• Supplementary file 2. Overview of models.

• Supplementary file 3. Models estimating R.

• Supplementary file 4. Forecast models.

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
