## [Decision Letter]

Thank you for submitting your work entitled "Improving epidemiological modeling coordination for Ebola and other public health emergencies" for peer review at *eLife*. Your submission has been favorably evaluated by Prabhat Jha (Senior editor), and three reviewers, one of whom (Mark Jit) is a member of our Board of Reviewing Editors. Another of the reviewers, Marco Ajelli, has also agreed to reveal their identity.

The reviewers have discussed the reviews with one another and the Reviewing editor has drafted this decision to help you prepare a revised submission.

The reviewers all agreed that this is a timely, comprehensive and useful review of modelling studies that have described and/or forecasted the Ebola epidemic in West Africa. We all enjoyed reading the paper, and agreed that it made some important points about good modelling practice in general. In particular, we agree that data sharing and complete access to best available epidemiological data (especially patient databases) for all modelling groups is of paramount importance in the case of future outbreaks of infectious diseases.

We believe that the way the information and discussion was presented could be improved, and by doing so would enhance the influence that the paper may have. In particular:

1) More detail could be provided on how Ebola modelling was actually used to support decision making both internationally and for individual national governments. Many of the recommendations in the Discussion are predicated on the premise that the modelling results were actually useful to inform policy (or if they weren't, could be made useful by improving the modelling process). Some historical discussion on what happened in practice would help here. Currently only a single US-centric reference is provided.

2) The paper needs a proper Materials and methods section providing details about the search terms/strategy, inclusion/exclusion criteria and information extracted from the papers. Given that this is a review, compliance with PRISMA guidelines would normally be expected. If a non-standard approach to searching was used, this needs to be justified and the method explained.

3) Each section of the paper seems to contain a mixture of (i) data directly extracted from the reviewed papers, (ii) direct inferences from these data, (iii) more general discussion points around the inferences and (iv) broader discussion (e.g. recommendations for standardisation and funding for future outbreak response). While each section is helpful, the way the paper is written makes it difficult to distinguish the provenance of the different points made. For instance, there are statements like "Modelers welcomed the release of these data, but expended considerable effort curating them for analysis", "accurate epidemic forecasting for Ebola proved to be quite challenging, especially for long time horizons, because the behavior of people infected and at risk was unpredictable", "incomplete information from the field can make it difficult to determine if failure to end the epidemic reflects sub-optimal implementation of effective measures, or over-optimistic assumptions about their effectiveness" where it is unclear whether these are statements drawn from the reviewed papers, or based on the authors' own experiences.

Using a traditional separation into Results and Discussion may also help this as most scientists are used to the idea of drawing immediate inferences from data, and then suggesting broader implications of these inferences. The Discussion section may be lengthy and structured if necessary. The broader discussion around funding etc. which is not really an extrapolation from the data (however welcome the points may be) should probably be given a section of its own.

4) Table 2 shows a very large variation in the baseline scenario forecast, spanning orders of magnitude. The authors seem to suggest that this stems from different assumptions used about epidemic growth forecasting e.g. random mixing, incidence decay or spatial/sociodemographic structuring. It would be useful if you could explicitly structure the presentation of results to show whether these different assumptions had an effect on the outbreak size. This should at minimum be done in the visual presentation but if some sort of meta-analysis/meta-regression could help that would be even better.

5) Additional information about the spatial structure (if any) of the models reviewed would be useful. Spatial structure in these models ranges from homogeneous (non-spatial models) to detailed agent-based models such as the one developed by Merler et al. Lancet ID.

6) Some models employed single streams of data (e.g., all case counts by country) while other models employed multiple data streams (e.g., all cases and cases among health-care workers such as the multi-type branching process model by Drake et al.). This would be useful to highlight.

7) The paper makes the point that there are differences between the results of various modelling groups because of different methodologies used. However, in some cases modelling methodology has been improperly used, and this would be useful to highlight. In particular the following areas could be given attention:

a) Parameter estimation. In Box 1, you have underlined that the different models analyzed in this review have a remarkably different number of "unknown" parameters to be estimated. This represents a very important topic that could be given more space in this manuscript). For example:

A clear distinction should be made between estimated and imputed parameters. For instance the number of estimated parameters in Pandey et al. (2014) is 4, while the other parameters are somehow imputed from the literature. That said, still the 4 unknown parameters estimated in Pandey et al. (2014) are not univocally identifiable by using as input the dataset considered in that work.

In Rivers et al. (2014) about 10 (surely not all unequivocally identifiable) parameters have been estimated on the basis of case counts only.

b) Uncertainty. A good modelling practice (which is not always followed in the reviewed papers) is the assessment of the uncertainty due to i) the stochasticity of the infection transmission process, and ii) the uncertainty in parameter estimates. In particular, as regards the uncertainty in parameter estimates, methods providing estimates of the parameters distribution (e.g., MCMC) should be preferred with respect to methods providing point estimate of the parameter values only (e.g., MSE, ML). This would avoid (or at least limit) the spread in the community of biased findings never confirmed on the field or by subsequent studies (e.g., the estimate that the main driver of the Ebola epidemic was the transmission during traditional burials found by Pandey et al. (2014) which was based on a point estimate made through a MSE fitting procedure).

c) Model validation. Another major point is that most (though not all) the reviewed modelling studies lack in model validation against other epidemiologically relevant quantities (e.g., ETU admissions, number of individuals included in contact tracing). Validation against other quantities is a key indicator to understand whether the model is able to explain the reason of an observed pattern of infection spread or it is just able to reproduce case counts "by chance". In fact, although it is welcome to show that the model is in agreement with the next few incidence points after stopping model calibration, modelers should not limit model validation to that.

8) It would also be good to clarify that the reviewed modelling studies can be divided in two categories: the ones aimed at explaining the observed pattern, for instance by accounting for actual data on intervention measures, and those that do not provide any explanation for the observed pattern (e.g., the model in Fisman, Khoo and Tuite (2014)). In fact, the point here is not to fit some data points, but to understand the mechanisms behind the observed infection spread as their knowledge is key for predictions and preparedness.

9) The importance for modelling groups of having access to online epidemiological data in an easy to export format (e.g., CSV file) could be highlighted. In fact, the Ministries of Health of the three most affected countries provided PDF files only, some of which were images, thus implying remarkable efforts by modelers to readily use such datasets. Moreover, the case of Guinea was even worse as, at the beginning of the epidemic, there was not an official website of the Guinean Ministry of Health with their reports uploaded.

---

## [Author Response]

*1) More detail could be provided on how Ebola modelling was actually used to support decision making both internationally and for individual national governments. Many of the recommendations in the Discussion are predicated on the premise that the modelling results were actually useful to inform policy (or if they weren't, could be made useful by improving the modelling process). Some historical discussion on what happened in practice would help here. Currently only a single US-centric reference is provided.*

We recast the manuscript as more of a standard scoping literature review, with less emphasis on anecdotal evidence of the practical utility of modeling during the outbreak. We mostly have only our personal experiences to draw on for such evidence; the revision draws much more heavily on the peer-reviewed literature (please see response to Comment 2 below).

*2) The paper needs a proper Materials and methods section providing details about the search terms/strategy, inclusion/exclusion criteria and information extracted from the papers. Given that this is a review, compliance with PRISMA guidelines would normally be expected. If a non-standard approach to searching was used, this needs to be justified and the method explained.*

We revised the manuscript as a PRISMA-compliant literature review.

*3) Each section of the paper seems to contain a mixture of (i) data directly extracted from the reviewed papers, (ii) direct inferences from these data, (iii) more general discussion points around the inferences and (iv) broader discussion (e.g. recommendations for standardisation and funding for future outbreak response). While each section is helpful, the way the paper is written makes it difficult to distinguish the provenance of the different points made. For instance, there are statements like "Modelers welcomed the release of these data, but expended considerable effort curating them for analysis", "accurate epidemic forecasting for Ebola proved to be quite challenging, especially for long time horizons, because the behavior of people infected and at risk was unpredictable", "incomplete information from the field can make it difficult to determine if failure to end the epidemic reflects sub-optimal implementation of effective measures, or over-optimistic assumptions about their effectiveness" where it is unclear whether these are statements drawn from the reviewed papers, or based on the authors' own experiences. Using a traditional separation into Results and Discussion may also help this as most scientists are used to the idea of drawing immediate inferences from data, and then suggesting broader implications of these inferences. The Discussion section may be lengthy and structured if necessary. The broader discussion around funding etc. which is not really an extrapolation from the data (however welcome the points may be) should probably be given a section of its own.*

We agree that the original organization was non-standard and confusing. The revision is structured as a standard research paper, with Introduction, Materials and methods, Results, and Discussion. The broader implications and recommendations are included in the Discussion.

*4) Table 2 shows a very large variation in the baseline scenario forecast, spanning orders of magnitude. The authors seem to suggest that this stems from different assumptions used about epidemic growth forecasting e.g. random mixing, incidence decay or spatial/sociodemographic structuring. It would be useful if you could explicitly structure the presentation of results to show whether these different assumptions had an effect on the outbreak size. This should at minimum be done in the visual presentation but if some sort of meta-analysis/meta-regression could help that would be even better.*

We replaced the table with a figure that identifies several factors that may have influenced forecasting results in a clearer and more compelling way. Figure 6 shows the association between more accurate forecasts and earlier forecasts, shorter time horizons, and approaches that constrained predicted epidemic growth (for which we suggest the term “dampening”). We also performed a regression to quantify these effects, reported in the Results.

*5) Additional information about the spatial structure (if any) of the models reviewed would be useful. Spatial structure in these models ranges from homogeneous (non-spatial models) to detailed agent-based models such as the one developed by Merler et al. Lancet ID.*

We identified modeling studies that incorporated spatial data (see Results, subsection “Data sources”).

*6) Some models employed single streams of data (e.g., all case counts by country) while other models employed multiple data streams (e.g., all cases and cases among health-care workers such as the multi-type branching process model by Drake et al.). This would be useful to highlight.*

We added two paragraphs in the Results section on sources of data (Results, subsection “Data sources”) and hope this addresses the spirit of the excellent comment, which we interpret as a call for more thorough characterization of the types of model input data.

7) The paper makes the point that there are differences between the results of various modelling groups because of different methodologies used. However, in some cases modelling methodology has been improperly used, and this would be useful to highlight. In particular the following areas could be given attention: a) Parameter estimation. In Box 1, you have underlined that the different models analyzed in this review have a remarkably different number of "unknown" parameters to be estimated. This represents a very important topic that could be given more space in this manuscript). For example: A clear distinction should be made between estimated and imputed parameters. For instance the number of estimated parameters in Pandey et al. (2014) is 4, while the other parameters are somehow imputed from the literature. That said, still the 4 unknown parameters estimated in Pandey et al. (2014) are not univocally identifiable by using as input the dataset considered in that work. In Rivers et al. (2014) about 10 (surely not all unequivocally identifiable) parameters have been estimated on the basis of case counts only.

We provided the number of parameters for models estimating R, and distinguished between estimated and imputed values ([Supplementary-material SD3-data]).

*b) Uncertainty. A good modelling practice (which is not always followed in the reviewed papers) is the assessment of the uncertainty due to i) the stochasticity of the infection transmission process, and ii) the uncertainty in parameter estimates. In particular, as regards the uncertainty in parameter estimates, methods providing estimates of the parameters distribution (e.g., MCMC) should be preferred with respect to methods providing point estimate of the parameter values only (e.g., MSE, ML). This would avoid (or at least limit) the spread in the community of biased findings never confirmed on the field or by subsequent studies (e.g., the estimate that the main driver of the Ebola epidemic was the transmission during traditional burials found by Pandey et al. (2014) which was based on a point estimate made through a MSE fitting procedure).*

We added a detailed analysis of how use of cumulative versus disaggregated input data, and deterministic versus stochastic approaches to parameter estimation, may have influenced the reported uncertainty of R estimates (see Results, subsection “Modeling results: R and forecasts” as well as Figure 4 and Figure 5).

*c) Model validation. Another major point is that most (though not all) the reviewed modelling studies lack in model validation against other epidemiologically relevant quantities (e.g., ETU admissions, number of individuals included in contact tracing). Validation against other quantities is a key indicator to understand whether the model is able to explain the reason of an observed pattern of infection spread or it is just able to reproduce case counts "by chance". In fact, although it is welcome to show that the model is in agreement with the next few incidence points after stopping model calibration, modelers should not limit model validation to that.*

We added this excellent point to the Discussion (paragraph four):

“Validation of incidence forecasts against other relevant data, such as hospital admissions and contacts identified, also could provide evidence that the assumptions are sound.”

*8) It would also be good to clarify that the reviewed modelling studies can be divided in two categories: the ones aimed at explaining the observed pattern, for instance by accounting for actual data on intervention measures, and those that do not provide any explanation for the observed pattern (e.g., the model in Fisman, Khoo and Tuite (2014)). In fact, the point here is not to fit some data points, but to understand the mechanisms behind the observed infection spread as their knowledge is key for predictions and preparedness.*

We added this distinction to our characterization of the models (see Results, subsection “Overview of modeling applications”):

“Of the 125 models reported across the studies, 74% included mechanistic assumptions about disease transmission (e.g., compartmental, agent-based, or phylogenetic models), while 26% were purely phenomenological ([Supplementary-material SD2-data]).”

*9) The importance for modelling groups of having access to online epidemiological data in an easy to export format (e.g., CSV file) could be highlighted. In fact, the Ministries of Health of the three most affected countries provided PDF files only, some of which were images, thus implying remarkable efforts by modelers to readily use such datasets. Moreover, the case of Guinea was even worse as, at the beginning of the epidemic, there was not an official website of the Guinean Ministry of Health with their reports uploaded.*

We’re grateful for the chance to address this very important point, and added the recommendation to the Discussion (paragraph seven):

“Ideally, data should be made available online in machine-readable form to facilitate use in analyses. Modelers and other analysts expended enormous effort during the EVD epidemic transcribing data posted online in PDF documents.”